# Risky Business: Live Non-CITES Wildlife UK Imports and the Potential for Infectious Diseases

**DOI:** 10.3390/ani10091632

**Published:** 2020-09-11

**Authors:** Jennah Green, Emma Coulthard, John Norrey, David Megson, Neil D’Cruze

**Affiliations:** 1World Animal Protection, 222 Gray’s Inn Rd., London WC1X 8HB, UK; JennahGreen@worldanimalprotection.org; 2Ecology & Environment Research Centre, Department of Natural Sciences, Manchester Metropolitan University, Manchester M1 5GB, UK; E.Coulthard@mmu.ac.uk (E.C.); J.Norrey@mmu.ac.uk (J.N.); D.Megson@mmu.ac.uk (D.M.); 3Wildlife Conservation Research Unit, Department of Zoology, Recanati-Kaplan Centre, University of Oxford, Tubney House, Abingdon Road, Tubney, Abingdon OX13 5QL, UK

**Keywords:** human health, outbreaks, pandemics, wildlife trade, zoonotic disease, pet, exotic pet

## Abstract

**Simple Summary:**

The UK imports wild animals for commercial purposes from countries all across the world. We analyse a database of wildlife records from the UK’s Animal and Plant Health Agency (APHA) to summarise the volume and variety of non-CITES (Convention on International Trade in Endangered Species) listed wild animal imports over a recent 5-year period (2014–2018). We found that over 48 million individual animals were imported into the UK from 90 countries across nine global regions from 2014–2018. In terms of volume (semi-domesticated pigeons and game birds aside), amphibians were the most commonly imported group (73%), followed by reptiles (17%), mammals (4%), and birds (3%). The highest number of import records came from Europe and Africa, but the largest volume of animals came from North America and Asia. We review the potential for infectious diseases emerging from these vast and varied wildlife imports and discuss the potential threats they pose to public health. We also draw attention to an observed current lack of detail in the APHA database and suggest that better record keeping and reporting could help prevent and manage the introduction of infectious diseases.

**Abstract:**

International wildlife trade is recognised as a major transmission pathway for the movement of pathogenic organisms around the world. The UK is an active consumer of non-native live wild animals and is therefore subject to the risks posed by pathogen pollution from imported wildlife. Here, we characterise a key yet overlooked portion of the UK wildlife import market. We evaluate the trade in live non-CITES (Convention on International Trade in Endangered Species) wild terrestrial animals entering the UK over a 5-year period using data reported by the Animal and Plant Health Agency (APHA). Between 2014 and 2018, over 48 million individual animals, across five taxonomic classes and 24 taxonomic orders, were imported into the UK from 90 countries across nine global regions. The largest volumes of wild animals were imported from North America and Asia, and most of the import records were from Europe and Africa. Excluding Columbiformes (pigeons) and Galliformes (‘game birds’), amphibians were the most imported taxonomic class (73%), followed by reptiles (17%), mammals (4%), birds (3%), and arachnids (<1%). The records described herein provide insight into the scope and scale of non-CITES listed wildlife imported in to the UK. We describe the potential for pathogen pollution from these vast and varied wildlife imports and highlight the potential threats they pose to public health. We also draw attention to the lack of detail in the UK wildlife import records, which limits its ability to help prevent and manage introduced infectious diseases. We recommend that improved record keeping and reporting could prove beneficial in this regard.

## 1. Introduction

Billions of plants, animals, and their products are traded across the globe annually [1], including a marked amount of wildlife trade for commercial purposes. With an estimated economic value of US $300 billion per annum, wildlife trade is one of the largest and most complex commerce chains in the world [2]. In some developing regions, wildlife trade can also benefit local communities economically by providing local livelihood opportunities and financial incentives to protect wild spaces [3].

Despite these benefits, the increasing trade of wild animals to meet the growing demand from local and global markets has had significant negative impacts. A global assessment of biodiversity and ecosystem services in 2019 ranked exploitation among five key drivers of harmful ecosystem change [4]. Wildlife trade facilitates the introduction of species to new regions, where they compete with native species for resources, alter ecosystems, damage infrastructure and crops [5], and contribute to biodiversity loss, pathogen emergence, and lowered food security [2]. The extraction of wild animals from their habitats, and their transport through global markets, also involves major animal welfare, health, and conservation concerns [6].

One of the most significant threats posed by wildlife trade is the inadvertent movement of infectious agents across global boundaries [7]. The introduction of novel pathogens via wild animal hosts threatens public health, agricultural production, and biodiversity alike [5,8]. Wild animal species are thought to be the source of at least 70% of all zoonotic emerging infectious diseases [9]. More than 35 infectious diseases have emerged in humans since 1980 [7], the most pertinent example being COVID-19, the highly contagious coronavirus currently cast as a global health pandemic, which is thought to have been transmitted from animals to humans during live animal trade in a wet market [10].

Despite the current economic benefits, disease outbreaks associated with wildlife trade also periodically cause hundreds of billions of dollars of economic damage [11] and millions of human deaths [12], as illustrated by the recent global COVID-19 pandemic. Unless the nature of wildlife trade shifts considerably, the increasing rate of biotic exchange indicates there will be greater opportunities for pathogens to proliferate across the globe [8].

### 1.1. Global Wildlife Trade

Wildlife trade refers to activities relating to the harvest, transport, commercial exchange, and end use of wildlife and derived products [3]. Wild animal species are traded for a range of purposes including fashion, traditional medicine, food consumption, luxury goods, trophies, and exotic pets [13]. Each market is driven by unique economic, cultural, and societal motivations [2], which fluctuate over time. Socio-economic factors such as increasing access to wealth, the commercialisation of wild animals in the media, and the diversification of online marketplaces are thought to be contributing factors to the growing demand for non-domestic animal trade across the world [14,15].

The ownership of non-domesticated animals, or ‘exotic pets’, is a substantial part of the global trade in wildlife [16]. Wild species that are commonly traded as popular pets, such as snakes, turtles, and parrots, can carry diseases that cause infections in people [17]. The commercial trading of ‘exotic’ animals for the pet market has been identified as an important driving factor in the emergence of zoonotic diseases [16,18] and includes a range of taxonomic groups, including birds, reptiles, amphibians, and mammals [19].

As is the case for other types of wildlife trade, the trade in exotic pets can be legal, illegal, or a combination of both, depending on how a species is classified as it moves throughout the market chain [20]. Trade in live animals poses a risk to global human health regardless of legality, because pathogens transported on host organisms can be circulated regardless of legal conditions [12]. Illegal wildlife trade is characterised by “unlawful activities associated with the commercial exploitation and trade of wildlife specimens, either living organisms or harvested parts there-of” [3]. Illegal trade can be challenging to audit due to the amount of undetected and under-recorded trade that occurs by way of its illicit nature [6]. Here, we focus on legal trade records, which are readily available and immediately manageable [7]. We acknowledge that records of legal wildlife trade are not without fault. Fraudulent activity, inadequate record keeping, and the misidentification and mislabeling of species have all been attributed to accusations for wild animals traded globally [20].

### 1.2. Trade Data

The Convention on International Trade in Endangered Species (CITES) Trade Database hosts a large number of international wildlife trade records, but it only encompasses species that have been afforded different levels or types of protection from over-exploitation (https://trade.cites.org/). As such, species traded under the guidance of CITES represent a small fraction of all species that are traded internationally [19,20]. Broader records of wildlife trade are kept at regional and country-specific levels (for example the ‘United Nations Statistics Division for Comtrade’ for UN Member countries and the ‘Law Enforcement Management Information System’ (LEMIS) of the USA national government [2]). These datasets typically enable researchers to quantify the origin, purpose, amount, and diversity of taxonomic groups in commercial traffic, revealing the extraordinary magnitude of wildlife traded in each region [5].

The ‘Trade Control and Expert system’ (TRACES) is an online management tool for all non-CITES listed animals imported in to European Union (EU) countries. This web-based service is the system used for recording all trade of live animals, germplasm, and other animal-derived commodities into or through Member States territories. When animal-based consignments are exported to—or traded within—the EU, TRACES manages and records the trade route. Authorities post relevant documents online through TRACES, enabling border control authorities to check the consignments and their accompanying certificates to allow them to travel through the EU.

In the United Kingdom (UK), TRACES data can currently be accessed via submission of a Freedom of Information (FOI) request to the Animal and Plant Health Agency (APHA). APHA is an executive agency sponsored by the Department for Environment, Food, and Rural Affairs of the UK government. The aim of APHA is to “safeguard animal and plant health for the benefit of people, the environment and the economy” [21]. They are responsible for facilitating international trade in animals and products of animal origin into and out of the UK, as well as other activities tangentially related to animal disease surveillance and control, such as infectious disease research and the licensing and registration of wildlife.

### 1.3. This Study

Investigating and summarising wildlife trade data have value for scientists and policy makers because of the trade’s impact on global biodiversity and conservation, animal welfare, and infectious disease emergence [6]. Understanding wildlife trade requires both the identification of what species are being traded and where trade routes occur [22]. Such data are freely accessible on the TRACES database, but to our knowledge, no consolidation or assessment of this information for the UK exists.

Here, we aim to characterise the nature of the UK live terrestrial wildlife import market that is not currently regulated under CITES. We obtained data pertaining to all consignments of live non-domesticated animals (excluding CITES listed species and all fish) imported into the UK recorded on the TRACES database, via a Freedom of Information request to APHA. We evaluate the type and volume of species entering the UK over a 5-year period, with additional focus on the country of export for all species. Our aim was to provide an overview of the import data and to highlight some of the potential pathogens associated with taxa commonly imported in to the UK.

## 2. Materials and Methods

### 2.1. Data Collection

Data on the volume of non-CITES listed live wild vertebrates (excluding fish) imported into the UK between 2014 and 2018 from all other countries was obtained from the UK APHA via a Freedom of Information Act (FOI) request, which was received on 03.09.19 (Ref: ATIC1797). Available information regarding the lowest available taxonomic status (e.g., species, genus, family, order, or class), reported country of origin, country of export, source type (e.g., wild-caught, captive bred, ranched), and intended purpose (e.g., commercial, zoological, private collection) for all imports was also specifically requested during this process. Data on species considered as “domesticated” (i.e., animals that have been controlled and bred for human benefit over many generations, eventually resulting in changes to their genetic makeup and appearance; e.g., cat (*Felis catus*), dog (*Canis familiaris*) and cattle (*Bos taurus*)) were specifically excluded from this information request. Fish were also excluded due to the high number of individuals traded, which would alter the relative proportions of taxonomic groups and distort our data. Live fish can also be imported for aquaculture, which is a distinct market that would warrant a separate study of its own.

### 2.2. Data Management

The APHA dataset consisted of nine columns of information including “country of origin”, “country of destination”, “commodity”, “species”, “species list”, “species class list”, “species family list”, “declaration year”, and “total number of animals”. A total of 3100 records were provided in the original APHA dataset. Information relating to country of origin, country of destination, commodity, declaration year, and total number of animals were complete with no missing data. Any suspected duplicate records were removed from the APHA dataset by creating a unique code (using data entered for “country of origin”, “commodity”, “declaration year”, and “total number of animals”). This resulted in the removal of 1227 records.

Taxonomic related information was missing for a proportion of the remaining 1873 records (Species (26%, n = 795), Species list (30%, n = 920), Species Class List (12%, n = 361) and Species Family List (91%, n = 2833)) and the level of taxonomic detail was inconsistently applied across these data fields. Given the level of taxonomic uncertainty amongst the APHA records, only taxonomic class and order-level information (provided directly or added from information in the data column “species class list”) was used in our analyses. Instances where data on taxonomic status were still missing were classed as “Not recorded” (13.82%, n = 259). Trade records that referred to multiple taxa were classed separately as “unknown mixed imports” (19.81%, n = 371). All data relating to the taxa ‘Columbiformes’ (pigeons) and ‘Galliformes’ (‘game birds’) in the APHA database were separated out and analysed separately due to the high volume of this data (which accounted for 94% of the individual animals imported). It is likely that these trade records related to semi-domesticated bird species such as gamebirds, which were not a major focus of this study.

### 2.3. Statistical Analysis

All analysis was carried out in Excel, R, and RStudio (R Core Team, Vienna, Austria, 2020). We described the tabulated categorical data using descriptive statistics, including percentages, bar charts, circle plots, and heat maps. Chi-squared goodness of fit was used to investigate the distributions of these data across year, taxonomic group, and regions. A pairwise comparison after any significant chi-squared goodness-of-fit test was performed using the Package ‘RVAideMemoire’. Figures were produced using ggplot2 (New York, NY, USA) [23]. Trade diagrams were created using the package “circlize” (Heidelberg, Germany) [24].

## 3. Results

### 3.1. Total Volume Traded

The APHA records show that a total of 1873 individually identified wildlife import records were reported by the UK between January 2014 and December 2018. The number of annually declared wildlife import records remained relatively constant during the period examined, with a mean average of 374.6 (standard deviation (SD) = 17.6) wildlife records reported per year (Appendix A). These import records included a total of 48,929,569 individual animals. Thus, for the period 2014–2018, an annual mean average of 9,785,914 (SD = 1,449,534) non-CITES listed live animals were imported into the UK as recorded in the APHA database.

### 3.2. Columbiformes and Galliformes

In total, 93% (n = 45,518,548) of the total individual live animals traded between January 2014 and December 2018 comprised bird species belonging to the Galliforme (c. 99%, minimum n = 12,739,860) and Columbiforme (<1% minimum n = 3004). These data only accounted for 4% of the actual number of records (n = 61). Where data existed, references were made to “semi-domesticated” species (e.g., *Phasianus* spp. and *Perdix* spp.), which were presumed to be “gamebirds”. The quantities of these imports remained stable across the time period and were nearly always from France (c. 96.6%, minimum n = 43,968,013; Appendix B).

### 3.3. All Other Taxa

Of the remaining 7% of records (n = 1812) (excluding Columbiformes and Galliformes), data on taxonomic class was not available on 630 records (34.8%). However, a total of five different classes were present in the data (Figure 1). The most frequently traded was “Aves” (Birds) (23.8%, n = 432), followed by “Mammalia” (Mammals) (22.2%, n = 402), “Reptilia” (Reptiles) (14%, n = 253), “Amphibia” (Amphibians) (5.0%, n = 91), and lastly “Arachnida” (Arachnids) (0.2%, n = 4). A Chi-squared goodness of fit revealed that number of records was not evenly distributed across the classes of organisms traded (X^2^ = 894.74, df = 5, *p* < 0.001). A post-hoc pairwise comparison found a significant difference between all classes (*p* < 0.001) except Aves and Mammalia. The volume of wild animals traded was also unevenly distributed (X^2^ = 8174311, df = 5, *p* < 0.001). A post-hoc pairwise comparison found a significant difference between all classes (*p* < 0.001).

The data comprised a range of taxonomic orders. Aside from Columbiformes and Galliformes, a total of 24 orders were recorded in the dataset; however, Sauria and Serpentes were combined to form Squamata, incorporating 23 different taxonomic orders. Several records referred to an order grouping of “Other birds” in the dataset. Despite a lack of clarity, this was left in due to a lack of further detail on these records. The taxonomic order most frequently traded was the ‘Psittaciformes’ (parrots) (11.9%; n = 215), followed by “Other Birds” (10.5%; n = 191), “Artiodactyla” (even-toed ungulates) (9.4%; n = 171), “Squamata” (lizards and snakes) (8.3%; n = 151), and “Carnivora” (7.2%; n = 130) (Appendix C).

In relation to the taxonomic class and the actual volume of wild animals traded (Columbiformes and Galliformes aside), the highest number of individual non-CITES listed live animals entering the UK during this time period were amphibians (73.1%, n = 2,492,156), followed by reptiles (16.8%, n = 578,772), mammals (4.4%; n = 150,638), birds (2.9%; n = 99,111), and arachnids (0.03%; n = 1083). A further 2.6% (n = 89,261) did not have assigned taxonomic data (Figure 2).

In reference to specific taxa, Anura (frogs) were the most frequently reported taxonomic order (79.06%, min n = 2,492,155), followed by Squamata (snakes and lizards) (10.2%, n = 348,151), Testudinata (turtles) (6.6%, n = 224,237), Rodentia (rodents) (3.33%, n = 113,650), and Psittaciformes (parrots) (2.2%, n = 74,829). A further 76,512 individuals (2.24%) were recorded as mixed taxonomic imports, and 12,749 individuals (0.3%) were missing taxonomic data altogether (Appendix C).

A Chi-squared goodness of fit revealed that the number of wildlife records was not evenly distributed across order of organisms traded (X^2^ = 3363.4, df = 24, *p* < 0.001). A pairwise comparison test found significant differences across many order pairings (*p* < 0.05) with Anura, Artiodactyla, Carnivora, Aves, Psittaciformes, Squamata, and Testudinata being found to be higher than expected. The volume of individual wild animals traded was also unevenly distributed (X2 = 43,555,504, df = 24, *p* < 0.001). A pairwise comparison test found significant differences across many order pairings (*p* < 0.05) with Anura, Squamata, and Testudinata being found to be higher than expected.

### 3.4. Country of Export

According to the APHA database, the primary regions exporting non-CITES listed live wild animals into the UK were Europe and Africa based on number of records, and North America and Asia for the volume of individual wild animals imported (Figure 3). Between January 2014 and December 2018, non-CITES listed live wild animals were imported into the UK from nine different regions and from 90 different countries globally (Figure 4, Appendix D).

With regard to individually identified wildlife records for imports to the UK, the highest number reported in the APHA database were from the Czech Republic (24.5%, n = 444), followed by Germany (7.84%; n = 142), Belgium (5.63%; n = 102), France (4.8%; n = 87), Italy (4.8%; n = 86), the Netherlands (4.3%; n = 79), Slovakia (4.1%; n = 74), Spain (3.3%; n = 59), “other parts of the UK” (2.8%; n = 50), and the US (2.7%; n = 49) (Appendix D).

With regard to the volume of individual live wild animals imported into the UK, the top 10 exporting countries reported in the APHA database were the USA (68.0%; n = 2,320,343), Singapore (6.6%; n = 225,785), the Czech Republic (4.8%; n = 163,491), Ghana (2.6%; n = 87,028), Vietnam (2.3%; n = 77,234), Indonesia (2.0%; n = 68,231), Spain (1.8%; n = 61,117), Uzbekistan (1.8%; n = 59,524), Italy (1.6%; n = 53,037), and Hong Kong (1.1%; n = 36,069) (Figure 3, Appendix D).

### 3.5. Taxa of Concern

Between 2014 and 2018, amphibians were imported into the UK from six global regions (Figure 3) and 31 countries (Figure 4, Appendix D). The primary exporters of amphibian imports were the US (n = 7), Indonesia (n = 7), and Singapore (n = 6) based on number of records, and the US and Singapore based on the total number of specimens/animals imported. There are seven import records from the USA to the UK that involved 2,075,312 individual frogs, and six from Singapore to the UK that involved 225,773 individual frogs. All specimens appeared to be anurans (frogs); however, species-specific data was provided for only one record of 10 frogs, relating to the American bullfrog (*Lithobates catesbeianus*) imported from the USA.

Reptiles were imported from eight regions (Figure 3) and 56 countries (Figure 4, Appendix D). The primary exporters of reptile imports were the USA (n = 18), Ghana (n = 16), the Czech Republic (n = 12), and Germany (n = 12) based on the number of records, and the USA, Ghana, Uzbekistan, and Vietnam for the total number of specimens/animals imported. There are 18 import records from the USA to the UK that involved 240,293 individual reptiles and 16 from Ghana to the UK that involved 69,458 individual reptiles. Species-specific data were provided for one record of 100 freshwater red-eared slider turtles (*Trachemys scripta*) from the USA. There are 10 import records from Uzbekistan to the UK that involved 57,212 individual reptiles.

Mammals were imported from seven different regions (Figure 3) and 51 countries (Figure 4, Appendix D). The primary exporters of mammal imports were Germany (n = 36), the Netherlands (n = 33), the Czech Republic (n = 31), France (n = 31), and Spain, the Czech Republic, and Italy based on number of records. There are 22 import records from Spain to the UK that involved 60,140 individual mammals. There are 31 records from the Czech Republic to the UK that involved 35,737 individual mammals. A total of 81 Chiropterans (bats) were imported into the UK from Madagascar (n = 69), Guyana (n = 2), and the Czech Republic (n = 10). A total of 113,650 individual Rodentia (rodents) were imported from 19 different countries within Europe. A total of 943 individual Carnivora (carnivores) were imported from 47 countries across seven different regions.

Birds (excluding Colombiformes and Galliformes) were imported from seven regions (Figure 3) and 34 countries (Figure 4, Appendix D). The primary exporters of bird imports were the Czech Republic (n = 123), Belgium (n = 45), and Germany (n = 43) based on the number of records, and the Czech Republic, Slovakia, and Italy for number of specimens/animals imported. There are 123 import records from the Czech Republic to the UK that involved 75,829 individual birds and 24 from Slovakia to the UK that involved 6703 individual birds. A total of 74,829 individual Psittaciformes (parrots) were imported from 20 countries. A total of 18,369 “other birds” were imported from 29 countries across Africa, Asia, Europe, North America, Oceania, and South America.

## 4. Discussion

The UK is an active consumer of non-native live wild animals. The records described here provide an important insight into the scope and scale of non-CITES listed wildlife being imported into the UK. Over 45 million individual animals, across five taxonomic classes and 24 taxonomic orders, were imported into the UK during 2014–2018, with an annual mean average of 9,785,914 individual animals. These wild animals were exported from 90 different countries across nine different global regions. The international trade of wild animals in such quantities involves the risk of undesired pathogen pollution (i.e., the introduction of pathogenic viruses, bacteria, fungi and parasites into new environments) [2], of which the UK is no exception.

A number of zoonotic diseases have been identified in taxa that are commonly traded for the UK exotic pet market (Appendix E). Brief examples of the potential public health risks associated with the ongoing UK import of wild animals belonging to each of these taxonomic groups are summarised below.

### 4.1. Species

Mammals—The majority of emerging human diseases are thought to originate from mammals [25], and as such, their import represents a particularly prominent concern from a public health perspective [26]. According to the UK data records, at least 150,638 mammals were imported to the UK from 51 countries across seven global regions over the five-year study period. These records include taxonomic groups that are associated with serious emerging zoonotic infections. For example, bats (of which 81 individuals were imported into the UK from Madagascar, Guyana and the Czech Republic) have been implicated in the transmission of COVID-19, Ebola, Hendra, Marburg, SARS-coronavirus, Nipah, and various rabies-related viruses, all of which can cause currently untreatable diseases in people, often with high fatality rates [27,28,29].

The UK import records also include rodents (representing 75% of all UK mammal imports during the study period), which is a taxonomic group that also has the potential to transmit a number of diseases to humans (both directly (e.g., Hantavirus Pulmonary Syndrome, Leptospirosis, Plague, and Rat-bite Fever) and indirectly (e.g., Cowpox, Babesiosis, Lyme disease and Murine Typhus) [30,31]. A recent confirmed case of bubonic plague in Mongolia is thought to have originated from contact with a dead marmot [32]. The potential public health risks associated with the import of rodents were demonstrated by an outbreak of monkeypox in the USA during 2003 where a shipment of rodents originating from Ghana resulted in an outbreak affecting 72 people, which prompted domestic and international trade restrictions to control transmission [5].

Birds—Birds are also susceptible to many diseases common to humans [33], and there are previous examples of zoonotic disease emergence from birds imported to the UK. For example, in 2002, cockatoos (*Cacatua alba*, *Cacatua sulphurea citrinocristata* and *Cacatua sulphure*) entering the UK illegally from Singapore were found to be infected with psittacosis, which is a zoonotic respiratory infection that causes severe pneumonia in humans and has a fatality rate of up to 10–15% [34]. Wild caught birds are also natural reservoir hosts Avian *paramyxovirus 1* (which causes Newcastle disease in humans). Severe outbreaks of Newcastle disease have been recorded in parrots imported into the UK [35].

However, perhaps most notably from a public health perspective, in 2007, the UK government (along with the EU) posed a permanent ban on imports of wild caught live birds to prevent the spread of Avian Influenza following the discovery of H5N1-infected birds at a UK quarantine station [12,36]. Yet, this particular trade ban did not extend to birds of captive bred origin [12], which is demonstrated by the fact that UK import records report that at least 74,829 individual parrots were imported from 20 countries in the period 2014–2018. As such, additional bird-associated diseases in humans including histoplasmosis, Q fever, allergic alveolitis, salmonellosis, campylobacteriosis, and giardiasis [37] associated with captive sourced bird imports remains an ongoing potential public health concern in the UK.

Reptiles and Amphibians—Reptiles imported from tropical countries also have a high possibility of carrying potentially dangerous pathogens [38]. For example, it is thought that reptiles act as vectors for diseases that affect human health (such as Q fever and Lyme disease) and are responsible for some of the UK’s reported human Salmonella cases [39]. In 2011, public health concerns arose when *visceral pentastomiasis* (caused by *Armillifer armillatus* larvae) was diagnosed in a worker at a snake farm in The Gambia [40]. Similar snake farms (which export wild caught and captive reptiles on to the global exotic pet market) are in operation in Ghana [41], which exported 16,458 individual reptiles into the UK between 2014 and 2018.

In terms of trade volume, amphibians were the most frequently reported taxonomic group. In total, 2,492,155 amphibians were shipped from 31 countries across six different regions during the study period. From a public health perspective, amphibians have the potential to act as vectors for zoonotic disease transfer; for example, they can be responsible for human cases of infection with *Aeromonas* spp., *Mycobacterium marinum*, and *Salmonella* spp. [30], the latter of which cause severe infections in people, particularly children [39]. Although it is not a threat to human health, chytridiomycosis (a highly contagious fungal disease among amphibians thought to have contributed to the decline or extinction of at least 501 amphibian species across six continents [8]) was confirmed in a breeding population of bullfrogs in the UK in 2006 [42].

It is important to acknowledge that pathogenic agents can also be transported across geographical boundaries via the natural movement of wild animals, such as migratory species (e.g., some birds and bats). While there are some zoonotic diseases in humans that appear to have been tied to spillovers from migratory species, the majority have instead more likely resulted from human activities, particularly direct contact with wild animals during harvesting and handling, and increased proximity of humans and livestock to natural habitats [43]. Consequently, there are concerns that the amount of contact between wild animals and people throughout the trade chain could place public health at a greater risk of pathogen transmission than natural migratory processes.

### 4.2. Exporting Countries

Excluding the potentially semi-domesticated Colombiformes and Galliformes exported from Europe, the largest volume of wild animals came from North America and Asia, with the greatest number of records from Europe and Africa. In terms of volume, the majority of the wild animals imported into the UK were amphibians (73%), many of which were presumably intended for use as exotic pets. The aquatic nature of such species presents a particular challenge in terms of mitigating the risks of zoonotic disease introduction, as water is an effective transmission medium for pathogenic organisms, and survival outside the host may be for a significant duration [44]. In addition, considerations on how to responsibly dispose of any water used in shipments must also be taken into account when dealing in aquatic specimens [2].

Many of the wild animals that arrived into the UK were also exported from regions that have been identified as emerging disease hotspots—for example, in tropical lower latitude areas of Africa, Latin America, and Asia [9]. In particular, the UK import data show that several countries from these regions, such as El Salvador, Nicaragua, Cameroon, Madagascar, Singapore, and Indonesia, exported large volumes of reptiles and amphibians to the UK annually, although some also exported species from other taxonomic groups (see Figure 4 and Appendix F). Additionally, importing wild animals from other regions such as Europe is also not without risk. Zoonotic disease emergence is closely associated with regions of greater mammal biodiversity [45], and Europe has been identified as a hotspot for mammal zoonosis [25]. These are important considerations given that the UK imports large volumes of mammals from countries in mainland Europe such as the Czech Republic, Italy, and Spain.

Furthermore, while most current data implicate mammals as the primary zoonotic infection reservoir, a substantial reservoir of human pathogens also exist in non-mammalian animals in trade. The potential significance of disease emergence from non-mammalian reservoirs should not be discounted, and further research should address the under-representation of data for non-mammalian sources.

### 4.3. Current Biosecurity Measures

Despite its recognised role as a major transmission pathway for pathogenic organisms, the majority of regulatory oversight of the international wildlife trade (e.g., CITES) has no focus on preventing zoonotic disease introduction and no authority for biosecurity regulation [2,46]. Yet, research has shown that international trade agreements could be an effective way to manage zoonotic disease risk by limiting the amount of contact between humans and animals [47]. In lieu of establishing an additional international treaty specifically to address pathogen transmission, CITES is arguably well placed to adapt and incorporate disease spread via wildlife trade in its international remit [48].

Furthermore, in general, there is a lack of surveillance for key animal diseases in most countries, and minimal health monitoring systems exist surrounding the trade of some wild animals, heightening the potential risk for transboundary disease movement [49]. To address the problem of emerging infectious diseases arising from wild animal pathogens, the World Organisation for Animal Health (OIE) considers wild animal translocation as a particularly high-risk activity and advocates a prevention-led approach as part of a four-stage strategy [50]. In light of the high volume and diversity of non-native live wild animals being imported into the UK from across the globe, it is perhaps logical that prevention should also be prioritised with regard to pathogenic pollution.

Currently, biosecurity measures (including risk assessments, border controls, and risk-based post-border surveillance) are in place in the UK to ensure that live animals (and animal products) entering the UK from third (i.e., non-EU) countries are safe and meet the specific animal and public health required conditions for import [51]. With regard to live animals from third (i.e., non-EU) countries, imports must (1) come from an approved establishment in an approved country; (2) be accompanied by agreed animal and public health certification as appropriate; and (3) enter the EU at an approved Border Control Post where checks are carried out to ensure that the consignment meets import conditions [51]. These animal consignments can then only enter the UK once they have undergone veterinary checks with satisfactory results and a Common Veterinary Entry Document (CVED) has been issued clearing the consignment as acceptable for import [51]. For live animal consignments from EU Member States, at the time of writing, the animals must comply with an Intra-Community Trade Health Certificate and welfare transport assurances, they need to have undergone a veterinary health check prior to moving, and they may be subject to post movement checks at destination in the UK but do not need to enter via a Border Control Post.

### 4.4. Data Gaps

A lack of detail in the UK import records provided by the APHA limits the extent to which these data can be used to describe the scope, scale, and dynamics of this trade activity. For example, with regard to the taxonomic status of wildlife imports, a total of 76,512 individuals (2.24%) were recorded as ‘mixed taxonomic import’, and for 12,749 individuals (0.37%), no information was provided. This meant that it was not possible to determine total import quantities for some taxa, which is something of particular concern when trying to quantify the scale of potential zoonotic transmission threats. Furthermore, although taxonomic status was recorded by APHA using a ‘Commodity’ number code, these appeared to be limited to order level. Most (90%) of data records that did provide more detailed information described taxonomic status to the family level, but only 26% of records did so to the species level. In addition, the data record entries also do not include other important information such as the country of origin (rather, only country of export was included), the import purpose (e.g., whether the imports were intended for private ownership, commercial use, or zoological purposes) or the source of the animals (e.g., whether specimens were wild caught, ranched, captive born, or captive bred—all of which carry some degree of zoonotic disease risk [12]). One further point of note is that columns of the data were in some cases incorrectly titled (e.g., species order was under a column ‘species class list’), and some orders also appeared to have been grouped (e.g., ‘Other birds’) for no discernible reason. In addition to reducing the possible applications of the data, these inaccuracies may in turn lead to transcription errors or missing data due to lack of clarity.

### 4.5. Study Limitations

We intentionally chose to exclude CITES listed species from this analysis, as we specifically wanted to focus on non-listed species given concerns that they may be subject to less scrutiny (e.g., [2]). By omitting CITES listed taxa, there will undoubtedly be species and associated pathogenic organisms that pose a risk to public health, and/or are regularly imported to the UK, that we do no describe or discuss here. For example, our dataset does not include any primates because all primates are listed on CITES appendices. Yet, there are recent news reports expressing concern for the number of primates entering the UK and their potential to spread disease to humans [52]. This highlights that while this study provides an important summary of infectious pathogens potentially entering the UK, it is not a complete inventory of pathogen risk. Similarly, we did not attempt to quantify any illegal UK imports or any subsequent seizures of wildlife and remained focused on legal trade only. Nor did we focus on domesticated animal imports, or imports of fish, which can also be responsible for the transmission of human infectious diseases of zoonotic origin [53].

This study was further limited to UK import records provided by the APHA only and does not include a comparison with records held by exporting countries or the EU’s TRACES online management tool. Discrepancies between export and import data in trade databases occur e.g., [41] and may arise for a number of legitimate reasons [54]. More generally, it is also recognised that wildlife trade records can be error-prone, incomplete, and characterised by a degree of uncertainty (e.g., [55,56]). Therefore, there is a risk that our analysis may be an under- or overestimate and may have missed some pertinent details that would have been recorded elsewhere. Despite these limitations, to our knowledge, this is the most comprehensive report of non-CITES listed wildlife trade importation for this time period and of this scale.

Finally, it was beyond the remit of this study to include a detailed or systematic review of pathogen spillover potential. Future studies would benefit from including a meta-analysis or other form of quantitative review of pathogens associated with imported animals, or a specific inventory of pathogens associated with import taxa of concern.

### 4.6. Recommendations

Although biosecurity protocols can help to lower the risk of zoonotic disease introduction, effective surveillance and control is hindered by the wide variety of species involved and the often-complex natural history of zoonotic agents [57]. Consequently, there are concerns that the current global approach to surveillance is inadequate for some wildlife diseases [58], and the large volumes of wildlife imported likely render it challenging and costly to effectively screen all individuals, even if it were technically possible to do so. For example, the parasitic tapeworm *Echinococcus multilocularis* was incidentally spread to the UK via imported European beavers from Germany in 2007 [59]. The case was particularly concerning because the infected beaver had spent six months in quarantine [60], where biosecurity measures failed to identify this risk to public health. The infection was contained, but mainland Europe remains a high-risk region for potential future transmission of this parasite to the UK, which is the causative agent of alveolar echinococcosis (AE) [61].

It has been suggested that efforts to decrease contact between wild animals and people could prove to be the most practical and cost-effective approach in reducing the global human health threat posed by zoonotic diseases [7]. Consequently, from a policy perspective, trade bans have been proposed as a tool to help achieve these threat reduction goals [12]. There are concerns that wildlife trade bans could have unintended negative consequences on both wildlife and economically vulnerable communities if they are implemented in a manner that fails to also adequately address aspects such as consumer demand, enforcement, and livelihood dependencies [3]. However, the impacts of a global pandemic can also have far-reaching negative impacts on livelihoods and economies, which is a point brought into sharp relief by the coronavirus COVID-19 pandemic [62,63,64].

The potential merits and inferiorities of trade bans aside, given that the import of non-native wild animals remains an ongoing phenomenon, it is imperative that disease monitoring and surveillance efforts are maximised, with particular attention to species imported from regions identified as zoonotic ‘hotspots’ [9,45]. Our study serves to highlight that efforts to prevent, detect, and eradicate zoonotic diseases associated with the import of live non-native wildlife into the UK could be aided by more detailed and extensive record keeping by the AHPA. As such, we recommend the inclusion of additional data as currently recorded in the CITES Trade Database (e.g., taxonomic status (ideally to species level), country of origin, source, and purpose) and improved record keeping will ensure that data records are as complete and informative as possible.

## 5. Conclusions

Health issues at the human–animal–environment interface cannot be effectively addressed by one sector alone [65]. Collaboration across all sectors and disciplines responsible for health is required to address zoonotic diseases and other shared health threats at the human–animal–environment interface via an approach often referred to as “One Health” [65]. In the UK, the current regulatory environment comprises several different governmental departments and agencies that are responsible for different and overlapping aspects of zoonotic disease prevention and control (e.g., Public Health England’s National Infections Service (NIS), Public Health Wales’ Communicable Disease Surveillance Centre (CDSC), Defra’s APHA (which also provides services to the Welsh and Scottish Governments), to name but a few).

Although important and informative data are no doubt held elsewhere (e.g., under the auspices of other relevant government agencies), given the important role of wildlife trade as a transmission mechanism for zoonotic disease, the AHPA data records relating to non-CITES listed species represent a valuable source of information that could help efforts to address zoonotic diseases. Such improved record keeping and reporting would also aid ongoing efforts to better understand and limit the other unwanted negative impacts associated with this type of trade activity. For example, future areas of study could include associated negative animal welfare impacts (e.g., whether transport conditions meet minimum species-specific animal welfare criteria), conservation concerns (e.g., whether unsustainable numbers are being harvested from species with declining populations), and legal criteria (e.g., whether wildlife imports are fully compliant with relevant international and domestic trade legislation).

## Figures and Tables

**Figure 1 animals-10-01632-f001:**
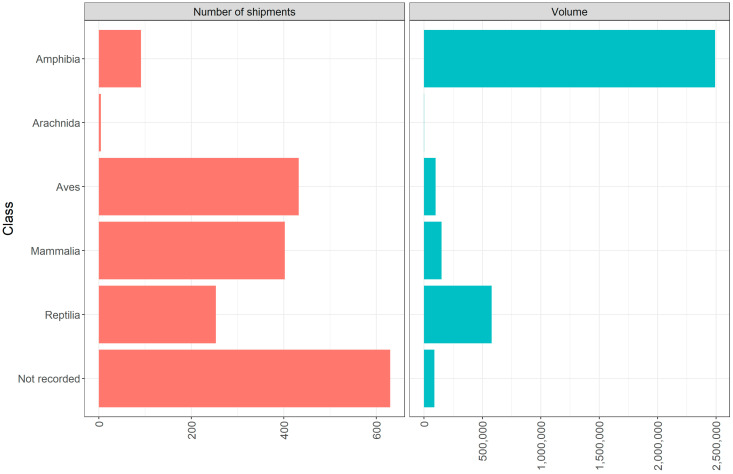
Classes of non-CITES (Convention on International Trade in Endangered Species) live wild animals traded 2014–2018 as per Animal and Plant Health Agency (APHA) database, with the number of records and total volume of individual wild animals. The ‘Aves’ data provided here exclude Columbiformes and Galliformes; see Appendix B.

**Figure 2 animals-10-01632-f002:**
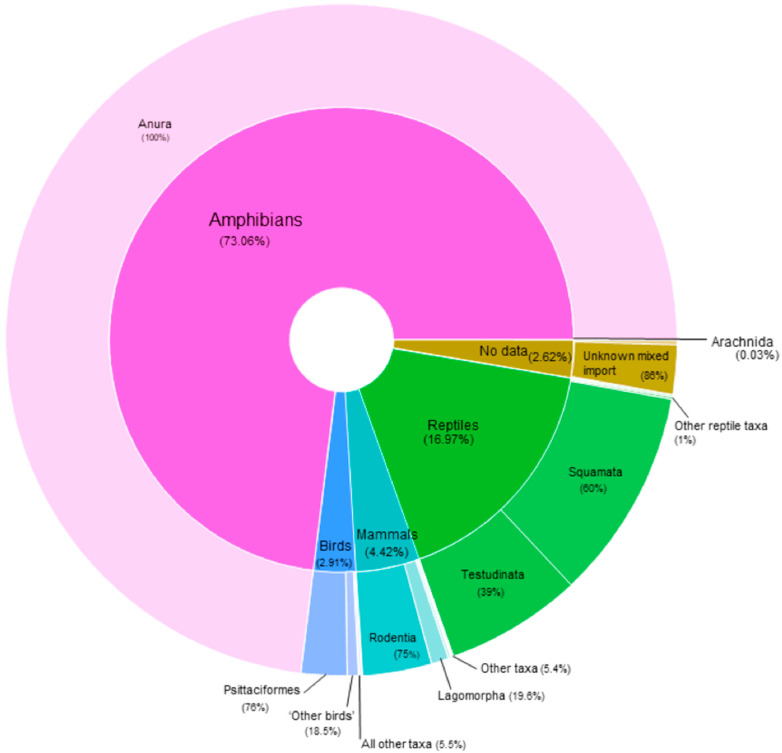
Percentage of taxonomic classes (inner) and respective orders (outer) (excluding Columbiformes and Galliformes) imported into the UK between 2014 and 2018 according to the APHA database.

**Figure 3 animals-10-01632-f003:**
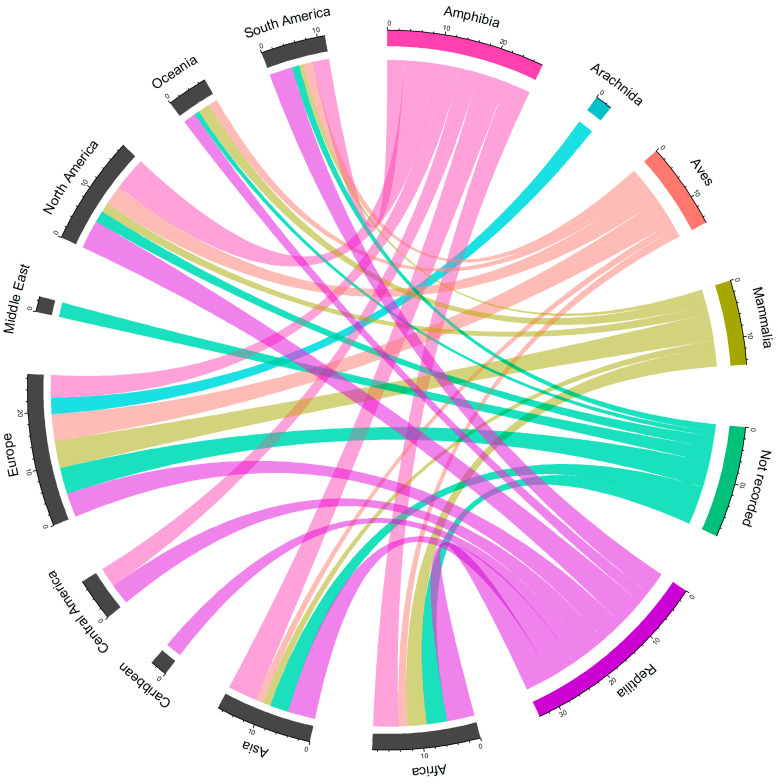
Circle plot representing the volume of non-CITES (Convention on International Trade in Endangered Species) listed live wild animals (excluding Columbiformes and Galliformes) imported from each of the different geographical regions according to each taxonomic class (log10).

**Figure 4 animals-10-01632-f004:**
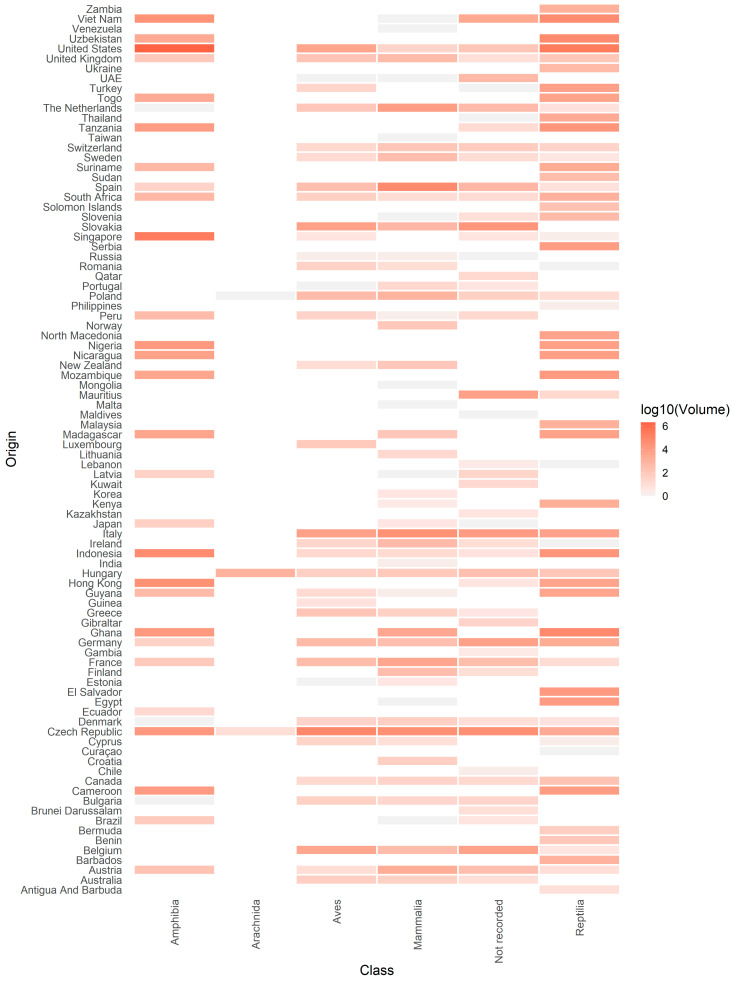
Heat map of the volume of non-CITES listed live wild animals (excluding Columbiformes and Galliformes) according to taxonomic class and country of export imported into the UK between 2014 and 2018. Data were log transformed (base-10 logarithm). Scale bar detonates the Log10 volume.

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
