# Peer review of "Risky Business: Live Non-CITES Wildlife UK Imports and the Potential for Infectious Diseases"

_animals, 2020, doi:10.3390/ani10091632_

Round 1

Reviewer 1 Report

The authors quantify non-CITES animal imports into the UK, finding nearly 10 million animals are imported every year. The authors drill down into the data, looking at the most frequent and most numerous taxa; the largest volume of which are amphibians. I find the subject area really worthwhile and, in the main, the questions asked are good. As a secondary aim the authors examine the risk of these imports for zoonotic spillover and as a public health issue. It is this part that I find rather weak and not well reviewed. The paper either needs a more comprehensive, and quantitative review of pathogens associated with imported taxa, or may even be improved by removing the focus on disease. I detail below my issues with this section. In addition, the stats would benefit from a different approach.

Broad comments: Given the introduction, I thought that this paper was going to present a meta-analysis or other form of quantitative review of pathogens associated with imported animals. Instead, a rather descriptive assessment occurs in the discussion. Given pathogen spillover is proposed as an aim the discussion seems an inappropriate place to introduce it. The aim needs a rethink, or the disease section needs a re-write. Given the brief and non-quantitative review the manuscript in its current form is a bit mismatched. Either the aim needs adjusting and the intro toning down a bit with respect to pathogen spillover over, or a more detailed spillover review is needed.

Currently the authors do not assess, in a detailed or a systematic way, pathogen spillover potential; this paper is much more a review focused on the actual numbers and volume and type of animals imported (which is a perfectly valid and interesting paper). If the pathogen data remains I take issue with the assessment of spillover into humans only. Zoonotic spillover from wildlife is indeed very high, but it is almost entirely from mammals (the authors do note in the discussion). Given that the majority of the animals imported are not mammals it is likely that there is very little public health risk from the millions of birds and amphibians imported into the UK. There is, however great risk of bringing in pathogens that will spill over into wildlife/domestic animals in the UK. I feel this is a point that is rather overlooked. Given the large number of amphibians chytrid is an obvious pathogen to discuss, but is only mentioned at the very end of the discussion.

In the disease section some of the pathogens mentioned are unlikely to pose a novel threat, as they are already endemic to the UK or require a vector not present, or present in very low numbers. For example, there are many migratory bird species in the UK that could easily import WNV from Europe and further afield, but to date it has not occurred. Therefore, I would suggest the authors clarify why imported birds would pose a greater potential risk for WNV import that the migratory birds. Some of the pathogens mentioned are vector-transmitted, or follow complex life-cycles such that even if they were found in an imported host would not spillover – this issue needs addressing. Other pathogens mentioned are already circulating in wildlife in the UK, and so imports pose very little novel threat. I suggest the section on disease is re-written and there is clear differentiation between spillover potential of entirely novel pathogens (i.e. could the next COVID-19 emerge), existing pathogens not currently circulating in the UK, and those already present. 

Analysis: Even though the volume of animals traded was significantly different across years it is clearly only 2017 that is causing the significance. A post hoc test to clarify this would be useful. The same comment goes for the Chi-squared looking at taxa differences. A GLM approach would be a better analysis throughout.

Figure 2 gives a nice illustration that the number of shipments and volume are really quite different. However, the number of shipments (%), volume (%) and mean volume seem to be superfluous columns showing the same pattern many times. I suggest removing these.

Minor points:

Line 82-83 - the authors state ‘pathogens do not care’. Please moderate the language here a bit to be less journalistic.

Can the authors clarify in the aims (c. line 123) that they propose to look at live animals only. Although the authors do clarify this later it is not clear at the moment where their aims are first presented.

The info presented in Figure 1 would be better presented in the main body of the text.

Line 208 it looks like you have a small typo or triple spaced between sentences.

Line 217 when you say ‘missing data’ do you mean no taxonomy assigned. Please clarify.

Line 218 do you really mean in reference to individual animals here? You go on to talk about taxa

I think it important in figure 3 to also clarify that you have excluded the game birds here, as you did in figure 4.

Figure 5 appears to be presented before it is cited, and needs some explanation in the text before it is presented. What is the scale on the right hand side?

Line 261 looks like you have a typo, with a double/triple space between the sentences.

The results are rather lengthy with every small detail explained, but this data is well presented in the figures. Do you need all of this info?

Sentence starting line 288 should be removed!

Discussion -  line 401, citing a news report weakens the discussion.  

Author Response

Thank you for your comments and feedback. Please see the attached document for details on how your suggestions have been addressed. Text and figures that have been added for the resubmitted version are recorded via tracked changes.

Reviewer 2 Report

This is an interesting article assessing the scope of the legal live wild animal trade in the UK with a mention of some of the zoonotic disease's threat posed by those animal movements. The recommendations are important and this article is worth publishing.
Please see my specific comments below:

418-420: authors state that "CITES is not currently specifically aimed at preventing zoonotic disease”. It is important to note that CITES is not at all aimed at preventing zoonotic diseases. CITES has no authority in terms of biosecurity regulation which fall under DEFRA in the UK. Enhanced coordination between regulatory authorities seems a more feasible approach.
429-439: This paragraph is not a recommendation but states measures already in place and should be moved higher up in the discussion to give the reader a perspective on the sanitary restrictions and measures already in place to minimise disease introduction into the UK.
For example, line 308, readers will understand that bats are entering the UK legally with no veterinary check and thus potentially introduce the diseases listed.
An analysis (even limited) of the risk posed by wildlife trade taken into account the specific biosecurity measures in place for each species and/or taxa should be included.

Minor: Remove line 289 290 - this doesn’t seem to belong to your text

Author Response

(The authors gave the same response as above.)

Reviewer 3 Report

The manuscript presented by Green et al. is very interesting and seem weel written, the movement of wildlife from countries is an important way to know and prevent the diseases transmission.

My major, mainly, concern is that this is in no way an "Article" which defined l as "Original research" manuscripts. The journal considers all original research manuscripts provided that the work reports scientifically sound experiments and provides a substantial amount of new information. In this investigation, Authors provided to analysed data just present in a database of APHA. In my opinion, there is no real research design, and there were a different category in which to put this article, such as "perspective," "opinion" or, at least, "Review".

Moreover there are two points necessary to expleain:

  • the health status of these imported animals was controlled by authority? How? When? This information should be added.
  • the mammalian species should be reports, due to they were moved for exampke for hunting acitvity, such as wild boar that represent an importart reservoir for different zoonitic diseases that could be infect hunters during slaughtering activity. May be this information should be added in a supplementary table.

Also, in the discussion Authors analysed the potential diseases risk associated to a specific imported animal species, this risk is potential because the aren't control during importation or it is an hypothetical perspection?

Also in the text are present several little mistake, i suggest to revised all manuscript to find them

Author Response

Thank you for your comments and feedback. We are happy for the editor to re-categorise our article if they also see this as more of a review than original research. We have reviewed the manuscript and amended all minor mistakes we could find in the text. We have included text about the authorities responsible for UK wildlife imports and their relevant bio-security measures. We cannot include the reasons for animal trade movements because ‘purpose’ is not recorded on the import database, but we have included an extra table in the appendix with specific examples of taxa of concern and their associated diseases.

Round 2

Reviewer 1 Report

The authors have addressed my concerns well and present a much improved manuscript. The issues in the text and presentation in figures has been addressed. The nature of the data was clarified, and the analytical approach is now clear and appropriate. This paper makes an interesting contribution to the literature.

Reviewer 3 Report

I appreciate the modification done by Authors.

My only one comment concernign the type of manuscript changes. I suggest to change from Article to Review or Perspective